# Extrusion Processing Modifications of a Dog Kibble at Large Scale Alter Levels of Starch Available to Animal Enzymatic Digestion

**DOI:** 10.3390/foods10112526

**Published:** 2021-10-21

**Authors:** Isabella Corsato Alvarenga, Lewis C. Keller, Christopher Waldy, Charles G. Aldrich

**Affiliations:** 1Department of Grain Science and Industry, Kansas State University, Manhattan, KS 66506, USA; isacorsato@ksu.edu (I.C.A.); lewkeller@gmail.com (L.C.K.); 2Pet Nutrition Center, Hill’s Pet Nutrition Inc., Topeka, KS 66617, USA; chris_waldy@hillspet.com

**Keywords:** extrusion, pet food, starch, gelatinization, resistant starch

## Abstract

The objective of the present work was to produce dog foods from a single recipe at three levels of resistant starch (RS). The low (LS), medium (MS), and high shear (HS) foods were produced on a single-screw extruder at target screw speeds of 250, 375 and 460 rpm, respectively, and with increasing in-barrel moisture as shear decreased. Post-production, kibble measurements and starch analyses were conducted. Kibble parameters were compared by ANOVA with significance noted at *p* < 0.05 with a single degree of freedom orthogonal contrasts for extrusion outputs, starch analyses, and viscosity (RVA). The MS and LS kibbles exiting the extruder were denser and less expanded (*p* < 0.05) than the HS treatment. Resistant starch, starch cook, and raw:cooked starch RVA AUC increased linearly as shear decreased. These results confirmed that lower mechanical energy processes led to decreased starch gelatinization and greater retention of in vitro RS.

## 1. Introduction

Extrusion cooking is the most common process used to produce pet foods worldwide [1]. Prior to extrusion, ingredients are ground and mixed. There is a preconditioning step before the dry mix enters the extruder barrel, where it is steam heated and further mixed to allow for hydration time. At the extruder barrel, thermal and mechanical energies are transferred to the dough, respectively, through water and steam additions and mechanical shear forces from screw components, which occur under pressure for a short time. Starches and other functional ingredients go through molecular changes that contribute to the formation of a viscoelastic material [2]. Under a moist environment and thermomechanical energy, starches gelatinize, paste, melt, and may fragment [2].

Corn is a starch ingredient commonly used in pet foods. Its initial gelatinization temperature was reported to range from 60 °C to 70 °C, and endothermic peak temperature to be close to 78 °C [2]. Gelatinization temperatures are affected by the amount of water available in the system, as well as heating rate and amount of starch damage [3]. Milling cereals with extensive shear will cause more starch damage, which changes the rapid-visco analyzer (RVA) profile to present an early elevation in viscosity due to starch cold swelling [4]. It is possible to control both milling and extrusion inputs in order to modify the extent of starch transformations, thereby modifying its utilization by dogs [5,6,7].

Although starch molecular transformations are important for matrix development and kibble expansion, these become readily available for enzymatic digestion in the animal’s small intestine, which in their extreme may not be desired nutritionally. Rapidly digestible starches may contribute to a sharp increase in blood glucose [8], whereas less cooked starches partially maintain their ordered structure and are either digested more slowly or are resistant to digestion [5,6,7]. Resistant starches (RS) are desirable because these bypass small intestine digestion and are fermented by saccharolytic bacteria in the colon, which promotes gut health [5,6,7].

The pet food industry is in constant search of innovation. In the present work, we propose to modify the nutrition of starch from corn to create a new product with higher amounts of RS, with the ultimate goal to benefit dog colonic health. Corn was chosen as the sole starch ingredient in the recipe for a few reasons. This cereal and its co-products are known to be well utilized by dogs, they are widely available for pet food companies in the US, and can promote desirable processing characteristics such as matrix development and expansion of the kibble [9]. Kibble physical attributes are important for consumer acceptance [10] and for sensory characteristics.

In a previous study, we explored the effects of changing processing parameters to create a surface response model to predict RS content in a dog food using a small-scale twin-screw extruder [11]. The treatment with the lowest extruder shaft speed (SS), highest in-barrel moisture (IBM), and greatest particle size yielded the greater RS. Thus, the objective of the present study was to use previous knowledge to produce diets with graded levels of RS through modification of processing parameters for future validation in an animal study.

## 2. Materials and Methods

### 2.1. Treatments and Extruder Settings

A single diet was formulated (Concept5^©^; CFC Tech Services Inc., Pierz, MN, USA) to meet the nutrient requirements for adult dogs at maintenance [12], with corn as its sole starch ingredient and no fiber ingredients (Table 1). Before extrusion corn was ground in a Jacobson 120-D portable hammermill (Carter Day International Inc., Minneapolis, MN, USA) using a 1.586 mm screen size, then mixed with the remained of the ground dry recipe and ground again through the same screen size into three batches of 1904.2 kg dry mix each. Although the same basal recipe was used to produce treatments, each differed in SS and IBM (Table 2) to target diets with three levels of RS and starch cook (low, medium, and high).

Each treatment was extruded in triplicate in a completely randomized design (CRD) experiment. Ration was preconditioned (Wenger Model 7 Dual Differential Conditioner DDC; Wenger Mfg., Sabetha, KS, USA) and extruded in a single screw extruder (Model X-115 Wenger Mfg., Sabetha, KS, USA) equipped with a 100 hp drive motor and an integrated operating system which provided real-time calculations for specific mechanical energy (SME), specific thermal energy (STE) and measurement of system temperatures and pressure readings. The extruder barrel was approximately 1.495 m long (11.41 cm screw diameter) with solid flight conveying screws and spiral liners, and a low shear configuration screw profile composed of 7 heads: head 1 was the inlet head with a tapered single flight, heads 2–5 were single flight conveying screws, head 6 was a double/solid flight, and 7 was a double/solid flight cone screw. The extruder die plate had 4 die holes of 0.82 cm diameter each. The treatments SS were set at either 250, 385, or 525 rpm, and IBM was targeted for moisture addition to being considered at low, medium and high levels with the expectation to produce diets with three levels of RS (Table 2) at high, medium, and low extruder shear. Other processing inputs included dry feed rate and preconditioner (PC) parameters (Table 2).

Post extrusion, kibbles from each replicate were dried in a forced convection dryer (Binder; Model FP240; Bohemia, NY, USA), which was equipped with load cells and programmed to track loss in weight for moisture control (drier settings in Table A1). In sequence, dry kibbles of each replicate were enrobed (using a Dinnessen without vacuum) separately with fat and dry topical flavor, and coated samples were collected then composited, mixed, and packaged for a subsequent feeding study.

### 2.2. Data Collection

Once the process achieved a steady state of moisture and extrudate temperature inside the die, samples out of the extruder were collected at three time points during each replicate production (at 0, 10, and 20 min). Each treatment transition lasted approximately 20 min. Photographs of the extruder panel were taken at the same time of sample collection, and each replicate data was averaged to provide representative processing parameters. Bulk density (g/L; off the extruder and off the dryer) and flow rate (kg/s) were measured at the same time as sample collection. Post extrusion, a fraction of each subsample was weighed, combined, mixed, and ground prior to laboratory analysis. This allowed for representative data per replicate.

### 2.3. Sample Analyses

Wet finished kibbles collected after the extruder die were subjected to RVA as described by [11]. Data were reported as the area under the curve (AUC) of cooked starch (0.4–6.0 min), raw starch (6.1–14.0 min), and high molecular weight (MW) starch (setback viscosity; 14.1–23 min). Starch cook was analyzed by the glucoamylase procedure described by [13]. Resistant starch, as well as rapidly, slowly, and total digestible starch were measured on the dry ground raw recipe (K-DSTRS kit Megazyme Inc., Wicklow, Ireland). In brief, 1 g extruded sample (ground at 0.5 mm) was weighed into glass flasks in duplicate and incubated at 37 °C with a buffer at pH 6.0 and constant stirring at 170 rpm. A mixture of 4 KU pancreatic α-amylase and 1.7 KU amyloglucosidase was added to each flask (in 41 mL suspension). Free glucose was measured at 510 nM wavelength on a plate reader (Gen5TM, Biotek^®^ Instruments, Inc. Winooski, VT, USA) to calculate the amount of starch digested. Rapidly and slowly digestible starch were the starch fractions digested after 20 min and 120 min of incubation, respectively. Total digestible starch was the total starch digested within 4 h of incubation, and RS was the undigested fraction after 4h. Insoluble and soluble dietary fibers (IDF and SDF) were determined on extruded replicates using an enzymatic kit (K-RINTDF; Megazyme Inc., Wicklow, Ireland).

Texture analysis was performed on 20 dried kibbles per replicate using a 25 mm cylindrical compression probe equipped with a 50 kg load-cell (TA-XT2; Texture Technology Corp., Scarsdale, NJ, USA) at 50% strain level. Prior to the analysis, kibbles were equilibrated in an oven at 40 °C overnight. The endpoints measured were kibble toughness (kgxmm) and hardness (kg) which was considered the highest significant fracture force per compression. The diameter and length of 15 wet and dry kibbles were measured twice with a digital caliper and averaged, then these were weighed on an analytical balance (Ohaus, Explorer: E1RW60, OHAUS, Parsippany, NJ) to calculate kibble density and expansion indices. Kibble volumetric expansion index (VEI) was calculated on wet kibbles according to [14]: VEI = (ρd × (1 − Md))/(ρe × (1 − Me))(1)
where ρd = extrudate density inside the die; Md = moisture content of the extrudate in the die; ρe = apparent density of the wet kibble; and Me = moisture content of the wet kibble. Moisture content inside the die (Md) was estimated to equal IBM. Steam loss was estimated according to [15] and subtracted from IBM to calculate moisture content of the wet extrudate after exiting the die (Me). The density of the kibbles inside the die (ρd) was estimated using the technique of [16] based on proximate composition and temperature inside the die. Sectional expansion index (SEI) was calculated as cm^2^e/cm^2^d, where cm^2^e is the squared kibble diameter, and cm^2^d is the squared die diameter. Lastly, the longitudinal expansion index (LEI) was calculated as a function of VEI divided by SEI.

To provide a visual representation of kibble expansion, light microscopy images were captured at 15× magnification using an SZH-ILLK Olympus Illumination Base (Olympus Optical Co., Ltd., Tokyo, Japan) equipped with a digital camera.

### 2.4. Statistical Analysis

The experiment was conducted as a complete randomized design (CRD). Single degree of freedom orthogonal contrasts for extrusion outputs, starch analyses, and viscosity (measured by RVA) were performed using the generalized linear mixed model (GLIMMIX) procedure from statistical analysis software (SAS v 9.4; Cary, NC, USA), and linear (L) and quadratic (Q) relationships were considered significant at a *p* < 0.05. Analysis of variance of kibble measurements and expansion indices were performed by the GLIMMIX procedure (SAS v 9.4; Cary, NC, USA) with replicate nested within the diet, and the means were considered significantly different at a *p* < 0.05. Multiple testing was adjusted by the Tukey–Kramer post hoc test.

## 3. Results

Each experimental diet was produced three times on a single day in a randomized order as follows: medium shear (MS), high shear (HS), low shear (LS), HS, MS, LS, MS, HS, and LS. The HS diet was produced with the highest extruder shaft speed (SS), while the LS was extruded with the lowest SS, and MS was intermediate (Table 2). The dry feed rate was initially set at 817 kg/h (Table 2), but production had to be stopped at the beginning of the first LS replicate due to kibble clumping. Once production was restarted, the dry feed rate was set at 898 kg/h until the remaining replicates were extruded (Table 2). Pre-conditioner (PC) shaft speed and steam were kept constant throughout extrusion, and PC water was modified to be the lowest in the HS and highest in the LS treatment. However, when the operator attempted to increase PC water in the LS above 23%, the wet kibbles exiting the extruder started to agglomerate and the added water application had to be decreased. As a result, the PC moisture and IBM increased from the high to medium shear process and plateaued from medium to low shear (*p* < 0.0001; Table 3).

The pre-conditioner temperature was targeted to remain constant across treatments at 88 °C (Table 3). Mass flow rate had a slight linear increase (*p* < 0.05) due to the PC water additions in the MS and LS treatments. Extrusion motor load and SME had a quadratic decrease (*p* < 0.05) from HS to LS driven by the decrease in SS and increase in IBM in each treatment level. Although STE was not significant among extrudates, the sum of total specific energy (TSE) decreased in a linear fashion (*p* < 0.05). As a result of extrusion inputs, both kibble wet and dry bulk densities increased linearly (*p* < 0.05) and moisture lost at the drier was greater in the treatments with less intensive cooking (MS and LS; Table 3).

The HS wet kibble was lighter (*p* < 0.05) and more expanded volumetrically (VEI) and longitudinally (*p* < 0.05) than the MS and LS treatments, which had similar expansion indices to each other (Table 4). Upon drying, both volume and kibble density were not different among treatments, but the LS kibbles had a tendency (*p* < 0.1) to be harder and tougher than the other treatments (Table 4).

Sectional and longitudinal microscopic imaging of kibbles did not show obvious differences in expansion or cell structure (Figure 1), which corroborate findings regarding expansion indices.

During the production of diets through this production scale extruder (Wenger model X115) the extrudate temperature inside the die was not recorded due to a system failure. To estimate steam flash-off at the die for kibble expansion calculation, die temperature was one of the factors needed. Hence, it was estimated by Equations (2) and (3): T_e_ = (Q_e_/(m_e_ × C_pe_)) + T_ref_(2)
where: Q_e_ = P_e_ × (motor load, %) × 36 + Q_se_ + Q_we_ + Q_p_(3)

T_e_; temperature of the product in the extruder just before the die (°C), Q_e_; total mechanical and thermal energy rate inside the extruder (kJ/h), m_ex_; total mass flow of water and dry material inside the extruder (kg/h), C_pe_; specific heat of the extrudate (kJ/kg·°C), T_ref_; the reference temperature (0 °C), P_e_; extruder motor power (kW), Q_se_; steam energy (kJ/h), Q_we_; water energy (kJ/h), Q_p_; preconditioner discharge energy (kJ/h), calculated by the weighted average of the mass fraction of carbohydrate, protein, ash, fat, and moisture inside the barrel.

Processing diets at various levels of thermomechanical energy led to different starch transformations. The RVA curve of the HS showed a more pronounced increase in viscosity from 0.4–6.0 min relative to the other treatments. This suggests that starch underwent more extensive chain scission and greater damage (Figure 2). Conversely, both the MS and LS kibbles may have had little to no cold swelling (little mechanically sheared starch). Moreover, the LS treatment had a greater final viscosity which preserved high MW starch that formed a gel upon cooling (Figure 2). The AUC which represents cold swollen starch decreased in a quadratic fashion (*p* < 0.05; Table 5) and the raw:cooked ratio (raw starch AUC divided by cold swollen starch AUC) increased linearly (*p* < 0.05) driven by the cold swollen starch AUC results.

Starch enzymatic assays confirmed the extent of starch that was available for digestion. Rapidly digested starch (RDS) had a marginal significance (*p* < 0.1) to decrease linearly from the HS to LS diet. Both the slowly digested starch (SDS) and RS increased linearly (*p* < 0.05) among treatments (Table 5). Conversely, cooked starch decreased linearly (*p* < 0.05) consistent with the thermomechanical energy that each treatment received. Fiber analysis was not different among treatments in any of the fiber fractions (Table 5).

## 4. Discussion

For a long time, pet food companies have targeted expanded, palatable, and highly digestible kibbles. There is an industry need for aesthetically pleasing and consistent croquettes (kibbles). This is usually achieved by fully cooking the starch under a high thermomechanical process known as food extrusion. While effectively cooked kibbles are sufficiently expanded [17], durable [18], aesthetically pleasing, and denature potential antinutritional compounds [19], their overall nutritional value may be compromised. Pet foods produced under high thermal and mechanical energies have been reported to result in vitamin losses [20], decreased availability of amino acids [21], among other nutrient changes [22,23].

Starches are primary ingredients in extruded foods due to their functional properties to bind particles, form a matrix, and assist with kibble expansion upon exiting the extruder die. Since this nutrient is not considered to be a dietary essential, it has been deemphasized regarding nutrition benefits until recently. Less cooked starches may partially retain native starch granules with greater RS compared to their highly cooked forms, which provides substrate for beneficial saccharolytic bacteria in the large intestine and act as a prebiotic [5,6,7]. Thus, starch may provide more benefit than merely an economical energy source.

It was the goal of the present study to produce diets with three levels of RS. This was achieved, although with less RS than anticipated. The corn used to produce the diets was ground through a hammermill using a 1.59 mm sieve size twice, which led to a smaller corn particle size that contributed to an increase in corn surface area (relative to mass) and starch gelatinization than what was targeted. [7] used a 2.00 mm sieve size to grind corn (once), and that along with decreased SME produced a diet with nearly 50% more RS than the LS food from the present study. Nevertheless, the present work was able to confirm that starch transformation measured with physical and enzymatic analyses differed according to processing parameters.

The modification of extrusion SME can have a large effect on starch gelatinization. Altering process settings can affect SME and consequently TSE. The preservation of partially gelatinized starches during the mild extrusion process has been shown to retain more RS [24] that acts as a prebiotic in both dogs [6] and cats [25]. In a recent study, [6] produced a high RS diet by decreasing extruder shaft speed and restricting die open area in order to increase the amount of energy transfer to the dough. At the opposite extreme of processing parameters (high shaft speed and no die opening restriction), a low RS food was produced. They were able to produce kibbles with a greater separation in TSE compared to the present study; wherein, the low RS had a TSE of 83.2 Wh/kg, while the high RS treatment was produced with nearly half that energy. Other authors have successfully modified the extrudate SME by solely altering the total die open area [5,7]. In the study by [6], the SME used to produce both low and high shear kibbles were lower than those obtained in the present study. This was likely a result of different extruder hardware and processing parameters.

The HS food in the present study had a 22% greater TSE compared to the LS, and this difference led to changes in kibble expansion, density, and extent of starch gelatinization. However, the change in total energy input was not equally distributed across treatments; wherein, TSE decreased by 15% from HS to MS, and only by 8% from MS to LS. Leading these last two treatments to be more closely related in starch cook levels and kibble characteristics. The kibbles produced at lower SME and higher IBM were denser, less expanded, and had a tendency to be tougher and harder than the food produced under high shear conditions. Other studies have also reported that a greater process moisture was positively correlated with hardness and negatively correlated with expansion [26]. In regards to SME, a higher screw speed will cause more starch shear, which may lower the melt viscosity and result in a more expanded and softer extrudate [17]. This corroborates findings from [27] regarding a high starch dry mix that produced harder extrudates under low screw speed and high moisture content.

Besides altering kibble characteristics, the differences in extrusion parameters in the present study also modified the starch digestion profile. The physical-viscosity method exemplified by the RVA methodology was performed to confirm starch transformations due to limitations in the RS and starch cook enzymatic assays. The main limitations of the RS and starch cook procedures are that starch digestion in vivo is dependent on physiological responses in the animal-like nutrient transit time, enzyme activity, intestine motility, presence of mucins, and hormones, which are not accounted for in vitro. Moreover, the digested starch from the starch cook procedure also includes some raw starch that can be digested due to the presence of pores and channels in corn and other cereals that can be sites for enzymatic adhesion [28,29]. In the RVA procedure, the addition of water while heating the starch-containing material disrupts amylose helices and crystallinity in raw starch which allows it to swell and paste [30]. The swelling and pasting allows for molecular entanglement and short-range interactions between starch molecules, which increase the system viscosity. The pasting phenomenon occurs due to amylose leaching, which increases the matrix viscosity and gels upon cooling, further increasing viscosity [30]. All these stages are measured in the RVA profile of a sample with temperature changes. The limitation of kibble RVA is that the composition of pet foods is not solely starch. For instance, the presence of starch–lipid interactions and protein agglomeration create additional responses which are not due to starch, which modify the intensity of peaks in each stage of the RVA curve and create jagged lines in the viscosity profile. For this reason, the total area under the curve corresponding to each starch transformation was reported rather than peak viscosities.

The RVA profile of the HS diet had a slight increase in cold temperature viscosity (below 25 °C), due to the high mechanical shear profile used to produce this treatment. This increase in cold viscosity reflects the presence of dextrins and the interactions among them [31]. Conversely, treatments MS and LS that were cooked with less mechanical energy resulted in lower initial cold swelling which translates into a greater presence of raw starches [31]. The final viscosity would be expected to be greater in the presence of more native raw starch [31]. However, the MS treatment had mixed outcomes: it behaved similar to a low shear profile for cold swollen (cooked) starch and raw starch viscosities, but more like the high shear food in regard to setback viscosity. When compared to the study that preceded this [11], the setback viscosity of the MS diet was closest to sample 16. This treatment was produced under the greatest SME amongst all the variables. Regardless, the present study confirmed that the MS food had an intermediate level of raw:cooked starch relative to the other treatments. This was also confirmed by the starch cook and resistant starch assays.

The last method explored in the present study to detect RS in dog kibbles was the total dietary fiber procedure (TDF). According to the Codex Alimentarius Commission (2010), dietary fiber includes carbohydrate polymers that are not hydrolyzed by the endogenous enzymes of humans. This may be extrapolated to other monogastric animals. Resistant starches from enzymatic digestion as part of the TDF procedure may be captured in the insoluble fiber component [32]. In the present study, the low concentration of RS associated with high variability in the TDF procedure did not result in this starch fraction creating a treatment difference.

## 5. Conclusions

The modifications in extrusion processing mechanical energy had an impact on kibble characteristics and starch transformation. The low shear (LS) and medium shear (MS) were more dense and less expanded than the high shear food (HS) and were more closely related due to their common IBM with little difference in the total specific energy imparted to the product. Physical and chemical starch analyses complemented and strengthened one another. The RVA profile indicated that high mechanical shear led to more starch damage than the other treatments. The starch raw: cooked ratio and RS of each food increased as mechanical energy decreased. The starch% cook assay corroborated the RS analysis. In conclusion, we were successful in producing diets with more RS by imparting less mechanical shear. The effects these diets have on overall dog health will be determined in a future study.

## Figures and Tables

**Figure 1 foods-10-02526-f001:**
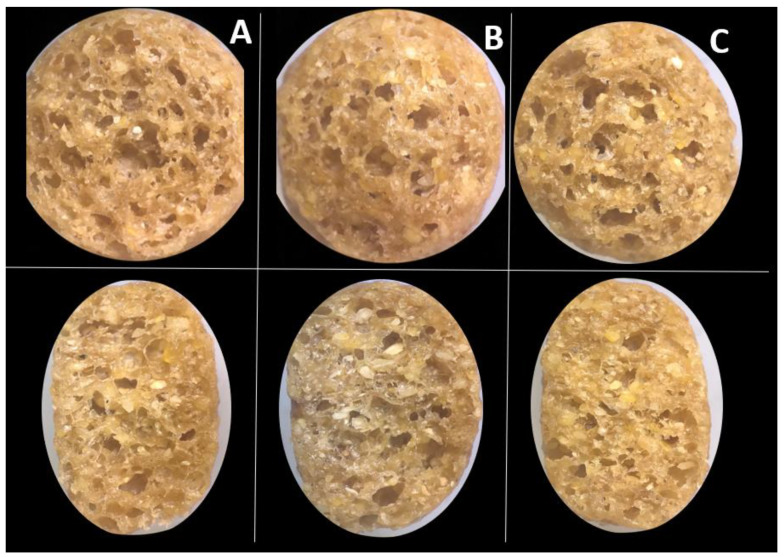
Light microscopy images of sectional (top row) and longitudinal (bottom row) cut of (**A**) high shear kibble; (**B**) medium shear kibble; and (**C**) low shear kibble.

**Figure 2 foods-10-02526-f002:**
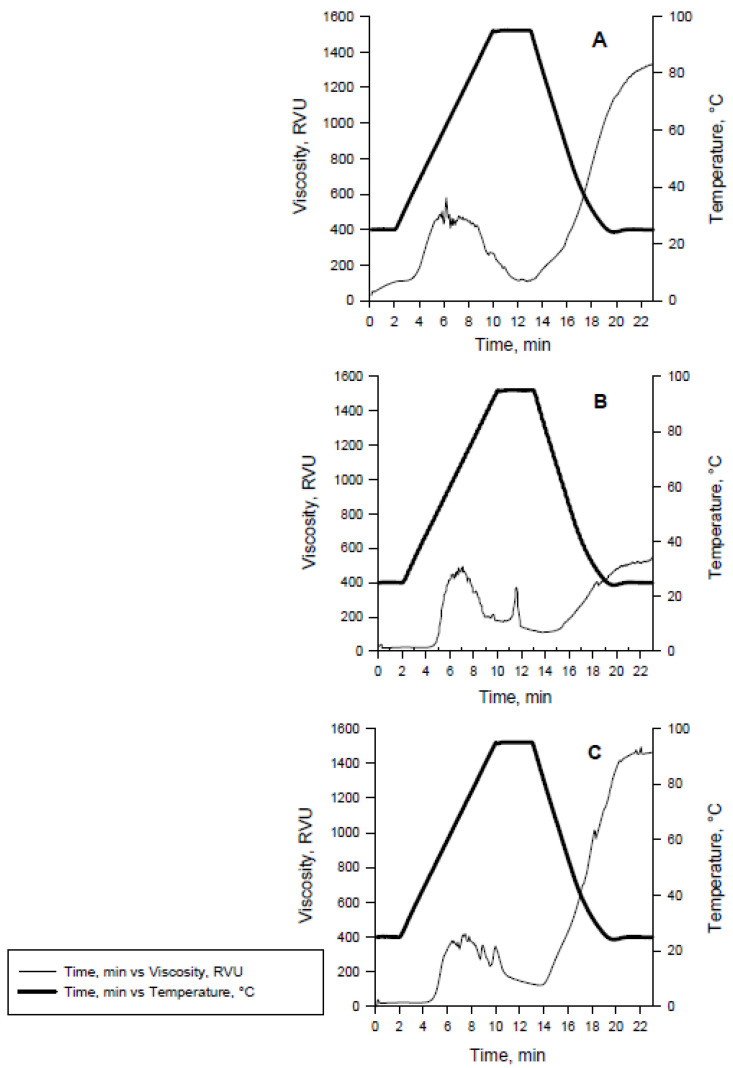
Rapid-visco analyzer (RVA) curve of the (**A**) high shear; (**B**) medium shear; (**C**) low shear food (average of three replicates each). Cooked starch AUC was considered from min 0.4–6.0, raw starch AUC was considered from min 6.1–14.0 min, and high Mw starch (setback viscosity) from 14.1–23 min.

**Table 1 foods-10-02526-t001:** Ingredient composition of the low, medium, and high RS experimental treatments, as-is basis.

Ingredient	Inclusion, %
Dry mix	
Whole yellow corn	65.4
Chicken meal	20.0
Potassium chloride	0.400
Vitamin premix	0.100
Lysine	0.100
Sodium chloride	0.100
Taurine	0.050
Mineral premix	0.100
Preconditioner	
Choline chloride, liquid, 70%	0.200
Lactic acid, blend 84%	1.50
Coating	
Choice white grease	8.40
Chicken, viscera and liver digest	3.00
Pork liver digest	0.500
Vitamin E, oil, 29%	0.100
Mineral premix	0.042

**Table 2 foods-10-02526-t002:** Input parameters and production sequence of a high shear (HS), medium shear (MS), and low shear (LS) dog food extruded on a single-screw extruder.^1^

Treatment	Production Sequence	Shaft Speed, rpm	Dry Feed Rate, kg/h	Estimated ^2^ PC, ^3^ RT, Min:Sec	PC Shaft Speed, rpm	PC Water, %	PC Water, kg/h	PC Steam, %	PC Steam, kg/h
MS	1	375	817	2:29	338	22	180	5.8	48
HS	2	458	817	2:34	337	14	110	5.6	46
LS	3	251	898	2:15	338	23	207	5.6	50
HS	4	457	898	2:20	339	14	121	5.6	50
MS	5	375	898	2:15	338	23	207	5.6	51
LS	6	251	898	2:15	338	23	207	5.6	50
MS	7	375	898	2:15	338	23	207	5.6	51
HS	8	457	898	2:20	338	14	121	5.6	51
LS	9	278	898	2:15	339	23	207	5.6	50

^1^ Dry mix bulk density was 668 g/L and moisture was 11.6%. Extruder water and steam were kept at 0% during all the runs, and knife speed was constant at 700 rpm. ^2^ PC = preconditioner. ^3^ RT = retention time.

**Table 3 foods-10-02526-t003:** Least square means and contrasts [linear (L); quadratic (Q)] for outputs from extrusion processing used to produce diets containing three levels of resistant starch.

Item	HS	MS	LS	SEM	L	Q
^1^ PC load, %	33.8	32.7	32.3	0.47	0.0729	0.5996
PC moisture, %	20.0	25.2	25.3	0.06	<0.0001	<0.0001
PC temperature, °C	88.9	88.0	87.2	0.35	0.0244	0.7549
Mass flow rate, kg/h	1184	1266	1308	31.7	0.0326	0.6142
Motor load, %	62.8	47.2	41.3	0.61	<0.0001	0.0017
^2^ SME, Wh/kg	39.5	27.9	23.6	0.85	<0.0001	0.0128
^3^ STE, Wh/kg	32.8	33.5	32.7	0.60	0.9748	0.3669
^4^ TSE, Wh/kg	72.2	61.3	56.3	1.36	0.0002	0.1194
^5^ IBM, %	25.8	31.2	31.3	0.09	<0.0001	<0.0001
Wet Bulk density, g/L	386	428	435	7.6	0.0039	0.1117
Dry Bulk density, g/L	296	324	338	6.2	0.0029	0.3615
Dry flow rate, kg/h	1101	1101	1136	28.1	0.4198	0.6349
Moisture loss at drier, %	6.98	13.04	13.16	0.078	<0.0001	<0.0001

^1^ PC = pre-conditioner; ^2^ SME = specific mechanical energy; ^3^ STE = specific thermal energy; ^4^ TSE = total specific energy; ^5^ IBM = in-barrel moisture, calculated.

**Table 4 foods-10-02526-t004:** Kibble parameters and texture analysis of X115 low, medium, and high RS diets (HS, MS, and LS, respectively).

Item	HS	MS	LS	SEM	*p*
Wet kibble					
Volume, cm^3^	1.541	1.257	1.260	0.0718	0.0497
Density, g/cm^3^	0.683 ^b^	0.866 ^a^	0.848 ^a^	0.0263	0.0049
VEI, cm^3^_kibble_/cm^3^_die_	1.097 ^a^	0.720 ^b^	0.728 ^b^	0.0291	0.0001
LEI, cm_kibble_/cm_die_	0.823 ^a^	0.611 ^b^	0.601 ^b^	0.0304	0.0034
SEI, cm^2^_kibble_/cm^2^_die_	1.353	1.188	1.232	0.0739	0.3336
Dry kibble					
Volume, cm^3^	1.56	1.57	1.55	0.061	0.9691
Density, g/cm^3^	0.527	0.540	0.554	0.0109	0.2770
Hardness, kg	8.37	8.13	9.96	0.534	0.0996
Toughness, kg × mm	1063	1101	1449	105.2	0.0758

^ab^ Least square means on the same row with unlike superscripts differ.

**Table 5 foods-10-02526-t005:** Least square means and contrasts [high vs. medium and low shear (T); linear (L); quadratic (Q)] for starch analyses from diets containing three levels of cooking.

Item	HS	MS	LS	SEM	*L*	*Q*
Viscosity (RVA)						
Cold swollen starch AUC, RVU	1120	402	330	72.61	0.0003	0.0109
Raw starch AUC, RVU	2206	1988	1996	110.2	0.2269	0.4327
^1^ Raw:cooked ratio	2.02	4.98	6.24	0.485	0.0009	0.2029
High M_w_ starch AUC, RVU	7281	3170	8411	1182.7	0.5243	0.0179
Starch analyses						
Rapidly digested starch, %	45.9	42.2	41.1	1.53	0.0686	0.5079
Slowly digested starch, %	2.02	2.98	6.41	1.072	0.0276	0.3832
Total digested starch, %	53.7	51.7	51.1	0.898	0.0909	0.5417
Resistant starch, %	0.650	0.940	1.057	0.0926	0.0210	0.4739
Total starch, %	54.3	52.6	52.2	0.94	0.1567	0.6050
Cooked starch, %	99.6	91.9	88.8	1.168	0.0006	0.1615
Fiber analysis						
TDF, %	2.51	2.64	3.03	0.159	0.0621	0.5410
IDF, %	1.46	1.67	1.70	0.197	0.4227	0.7102
SDF, %	1.052	0.969	1.326	0.1211	0.1598	0.1885

^1^ Calculated by dividing the area under the curve (AUC) between the raw starch and cold swollen starch RVA.

## Data Availability

Not applicable.

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
