# Peer review of "Extrusion Processing Modifications of a Dog Kibble at Large Scale Alter Levels of Starch Available to Animal Enzymatic Digestion"

_foods, 2021, doi:10.3390/foods10112526_

Round 1

Reviewer 1 Report

The manuscript entitled: "Extrusion processing modifications of a dog 2 kibble at large scale alter levels of starch available to 3 animal enzymatic digestion" (numbered by: foods-1381315)  can be treated as relevant to the field of FOODs journal. The paper presents interesting results related to influence  of operating parameters of single-screw extruder on the content of resistant starch (RS)  in animal foods (kibbles).

The title corresponds to content of the work, however, in my opinion, it should not contain numbers. 

Introduction

The Introduction section is very poor. The introduction does not refer in details to the aim of the work and research problem. The paper deals with the extrusion of corn pulp.  There is no information about conditions of this operation and problems with its implementation. At what temperature do starch transformations take place? On what devices is this process  implemented as standard? What was the reason that the Authors attempted this research? There is lack justification of aim of the work.

- The research problem is not clearly presented in this section too.

This part of work need to be expanded.

Materials and methods

The methods are properly adopted by authors and quite correctly described.

- The statistical significance "p-value" should be written in lowercase.

- There are many acronyms in the text, maybe i worth preparing a nomenclature.

Results and Discussion

The discussion and explanations of the results were made correctly on the basis of current research by other authors.

Figure 1 needs to be revised.

- a, b, c should be separated on figure 1 and properly described.

- Different scales of viscosity were used in the figures (-400 ???).

Conclusions

No specific conclusions follow from the presented work. This part of work should be expanded.

The selected topic is interesting. The work is properly documented and according to my opinion the paper need minor changes.

Author Response

Comments and Suggestions for Authors

The manuscript entitled: "Extrusion processing modifications of a dog 2 kibble at large scale alter levels of starch available to 3 animal enzymatic digestion" (numbered by: foods-1381315)  can be treated as relevant to the field of FOODs journal. The paper presents interesting results related to influence  of operating parameters of single-screw extruder on the content of resistant starch (RS)  in animal foods (kibbles).

Hello,

Thank you for taking the time to review our work. We appreciate your input and effort in helping us improve this manuscript. Please find replies to each of your comments below.

The title corresponds to content of the work, however, in my opinion, it should not contain numbers. Thank you. The only number the title has is “Part II”- if this is what you mean, we have removed it. The paper was assigned “part II” because it was a sequence of a preliminary extrusion work.

Introduction

The Introduction section is very poor. The introduction does not refer in details to the aim of the work and research problem. The paper deals with the extrusion of corn pulp.  There is no information about conditions of this operation and problems with its implementation. At what temperature do starch transformations take place? On what devices is this process  implemented as standard? What was the reason that the Authors attempted this research? There is lack justification of aim of the work.

- The research problem is not clearly presented in this section too.

This part of work need to be expanded.

Thank you for your comments. The introduction was expanded with two additional paragraphs where we stated more clearly the research problem and goal, as well as the initial and peak gelatinization temperatures of corn starch, and more RVA description.

Materials and methods

The methods are properly adopted by authors and quite correctly described.Thank you.

- The statistical significance "p-value" should be written in lowercase. The “P”s were replaced with “p.”

- There are many acronyms in the text, maybe i worth preparing a nomenclature. Thank you for the suggestion, we added an abbreviations list after the abstract.

Results and Discussion

The discussion and explanations of the results were made correctly on the basis of current research by other authors. Thank you.

Figure 1 needs to be revised.

- a, b, c should be separated on figure 1 and properly described.

- Different scales of viscosity were used in the figures (-400 ???).

We removed the titles above each figure and replaced these with A,B,C. These letters were assigned to RVA plots and described in the legend below. We do not understand the question about different scales of viscosity. They all have the same scale (from 0 to 1,600 RVU).

Conclusions

No specific conclusions follow from the presented work. This part of work should be expanded. Thank you. We added more detail on the findings regarding starch transformation in the conclusion section.

The selected topic is interesting. The work is properly documented and according to my opinion the paper need minor changes. Thank you! We appreciate your contribution to our work.

Reviewer 2 Report

Alvarenga et al. have used different shear forces to produce kibble with varied resistant starch, which is interesting. The physiochemical property and the functionality of the product were measured properly. The results were described in a clear way. However, I have a few major concerns and hope they can be useful for improvement of the manuscript.

Introduction

  1. The authors might want to add more contents regarding extrusion on production of less gelatinized starch on both human and animal foods. Also, how the shear force affects the product properties, such as hardness, density, etc.

Methods

2. More detail information should be given regarding the digestion of the starch. Enzyme to substrate ratio, reaction time, temperature, and the hydrolysate analysis,

Results & Discussion

3. RVA figure looks unprofessional. Suggest using more professional tools to put the three subfigures together for better comparison

4. In MS kibble RVA curve, why there is peak between 11-12 min, explanation for the appearance of the peak is required. Also, more discussion on the viscosity should be given. Viscosity depends on molecular entanglement and short-range interactions of starch molecules, which the authors might want to add.

5. I would suggest adding some microstructure images, at least a surface and cross-section images from light or confocal microscope, if have SEM, that will be even better.  

Author Response

Comments and Suggestions for Authors

Alvarenga et al. have used different shear forces to produce kibble with varied resistant starch, which is interesting. The physiochemical property and the functionality of the product were measured properly. The results were described in a clear way. However, I have a few major concerns and hope they can be useful for improvement of the manuscript. Thank you for your time dedicated in helping us improve our work. Please see below answers and changes we made on each point you suggested.

Introduction

  1. The authors might want to add more contents regarding extrusion on production of less gelatinized starch on both human and animal foods. Also, how the shear force affects the product properties, such as hardness, density, etc. Thank you for your comment. We have added two extra paragraphs in the Introduction to better explain starch transformations, and also added more information in the discussion specifically stating that more shear is expected to produce a softer and more expanded extrudate.

Methods

  1. More detail information should be given regarding the digestion of the starch. Enzyme to substrate ratio, reaction time, temperature, and the hydrolysate analysis.

Thank you. We have added a brief description on the starch fractions analysis under “2.3. Sample analyses”:  

“In brief, 1 g extruded samples (ground at 0.5 mm) were weighed into glass flasks in duplicate and incubated at 37°C with a buffer at pH 6.0 and constant stirring at 170 rpm. A mixture of 4 KU pancreatic α-amylase and 1.7 KU amyloglucosidase was added to each flask (in 41 mL suspension). Free glucose was measured at 510 nM wavelength on a plate reader (Gen5TM, Biotek® Instruments, Inc.  Winooski, VT, U.S.A.) to calculate the amount of starch digested. Rapidly and slowly digestible starch were the starch fractions digested after 20 min and 120 min of incubation, respectively. Total digestible starch was all the starch digest-ed within 4h of incubation, and RS was the undigested fraction after 4h.”

Results & Discussion

  1. RVA figure looks unprofessional. Suggest using more professional tools to put the three subfigures together for better comparison. Thank you for your comment. This was a similar comment to the other reviewer. Titles above the figures were removed and each plot was assigned a letter (A, B, C). The figure letters were described on the legend. We believe it looks more professional in the updated format.
  2. In MS kibble RVA curve, why there is peak between 11-12 min, explanation for the appearance of the peak is required. Also, more discussion on the viscosity should be given. Viscosity depends on molecular entanglement and short-range interactions of starch molecules, which the authors might want to add.

Thanks. We added “The swelling and pasting allows for molecular entanglement and short-range interactions between starch molecules, which increase the system viscosity.” On L382

Regarding the Unexpected RVA peak at 11-12 min, these unexplained peaks can be caused by things like protein agglomeration, starch lipid- or protein-lipid interactions. In this work we are only using RVA to give clues about what has happened to the starch biopolymers during extrusion, thus providing indicators of the effects of process energy intensity on starch which can be revealed by rapid visco-analysis. The existence of other peaks or jags are quite common with pet food formulations because they contain more protein and fat than direct-expanded formulations, which are high in starch and relatively low in protein and fat. We explained this happening in the discussion:

“The limitation of kibble RVA is that the composition of pet foods is not solely starch. For instance, the presence of starch-lipid interactions and protein agglomeration create additional responses which are not due to pure starch, and which modify the intensity of peaks in each stage of the RVA curve and create jagged lines in the viscosity profile. For this reason, total area under the curve corresponding to each starch transformation was reported rather than peak viscosities.

  1. I would suggest adding some microstructure images, at least a surface and cross-section images from light or confocal microscope, if have SEM, that will be even better.  

Although a visual representation of the kibbles cross section might be interesting, we believe it is not necessary because we already provided both chemical and physical proof of starch transformations. Measurements like bulk density and expansion also strengthened our findings. If the reviewers/editors  require kibble cross-section imaging, we would need more time to do the analysis. Thank you.

Round 2

Reviewer 2 Report

Comment 3:

Suggest the authors put A, B, C together in one figure for better comparison, this has not been done yet. The unprofessional figure means the no x-axis and y-axis, and the scale in x-axis should be re-made as they look crowding. All these should be improved. Origin or Sigma plot would help.

Comment 5:

Microstructure results will be helpful to direct reflects the change of the physiochemical properties. And it is more direct measurement. I do believe some microstructure (light, or CLSM or SEM) images are essential and meaningful.

Author Response

Comment 3:

Suggest the authors put A, B, C together in one figure for better comparison, this has not been done yet. The unprofessional figure means the no x-axis and y-axis, and the scale in x-axis should be re-made as they look crowding. All these should be improved. Origin or Sigma plot would help.

Thank you for your note. We have plotted all 3 RVA curves using Sigma Plot, and they look much more professional now. The 3 curves are separate, but all have the same scales and are all on the same page for an easy comparison. We did not plot all 3 curves on the same graph because the temperatures vary slightly, and the 3 plots can be well visualized with the new format.

Comment 5:

Microstructure results will be helpful to direct reflects the change of the physiochemical properties. And it is more direct measurement. I do believe some microstructure (light, or CLSM or SEM) images are essential and meaningful.

Thanks! 15 x microscopic images of kibbles were taken and added to the paper to provide a visual representation of our treatments. No evident differences could be seen, but this could be an interesting extra information for the reader.